# Disentangling Causal Effects from Sets of Interventions in the Presence of Unobserved Confounders

**Olivier Jeunen**[*]
Amazon
Edinburgh, UK

**Ciarán M. Gilligan-Lee**
Spotify & UCL
London, UK

**Rishabh Mehrotra**[*]
Sharechat
London, UK

**Mounia Lalmas**
Spotify
London, UK

## Abstract

The ability to answer causal questions is crucial in many domains, as causal inference allows one to understand the impact of interventions. In many applications, only a single intervention is possible at a given time. However, in some important areas, multiple interventions are concurrently applied. Disentangling the effects of single interventions from jointly applied interventions is a challenging task—especially as simultaneously applied interventions can interact. This problem is made harder still by unobserved confounders, which influence both treatments and outcome. We address this challenge by aiming to learn the effect of a single-intervention from both observational data and sets of interventions. We prove that this is not generally possible, but provide identification proofs demonstrating that it can be achieved under non-linear continuous structural causal models with additive, multivariate Gaussian noise—even when unobserved confounders are present. Importantly, we show how to incorporate observed covariates and learn heterogeneous treatment effects. Based on the identifiability proofs, we provide an algorithm that learns the causal model parameters by pooling data from different regimes and jointly maximizing the combined likelihood. The effectiveness of our method is empirically demonstrated on both synthetic and real-world data.

## 1 Introduction

The ability to answer causal questions is crucial in science, medicine, economics, and beyond, see [Gilligan-Lee, 2020] for a high-level overview. This is because causal inference allows one to understand the impact of interventions.[2] In many applications, only a single intervention is possible at a given time, or interventions are applied one after another in a sequential manner. However, in some important areas, multiple interventions are concurrently applied. For instance, in medicine, patients that possess many commodities may have to be simultaneously treated with multiple prescriptions; in computational advertising, people may be targeted by multiple concurrent campaigns; and in dietetics, the nutritional content of meals can be considered a joint intervention from which we wish to learn the effects of individual nutritional components.

Disentangling the effects of single interventions from jointly applied interventions is a challenging task—especially as simultaneously applied interventions can interact, leading to consequences not seen when considering single interventions separately. This problem is made harder still by the possible presence of unobserved confounders, which influence both treatments and outcome. This paper addresses this challenge, by aiming to learn the effect of a single-intervention from both observational

---

[*]Work done while author was at Spotify.

[2]Causal inference also allows one to ask and answer *counterfactual* questions, see [Perov et al., 2020] and [Vlontzos et al., 2022].

36th Conference on Neural Information Processing Systems (NeurIPS 2022).

data and sets of interventions. We prove that this is not generally possible, but provide identification proofs demonstrating it can be achieved in certain classes of non-linear continuous causal models with additive multivariate Gaussian noise—even in the presence of unobserved confounders. This reasonably weak additive noise assumption is prevalent in the causal inference and discovery literature [Rolland et al., 2022, Saengkyongam and Silva, 2020, Kilbertus et al., 2020]. Importantly, we show how to incorporate observed covariates, which can be high-dimensional, and hence learn heterogeneous treatment effects for single-interventions. Our main contributions are:

1. A proof that without restrictions on the causal model, single-intervention effects cannot be identified from observations and joint-interventions. (§3.1,3.2)

2. Proofs that single-interventions *can* be identified from observations and joint-interventions when the causal model belongs to certain (but not all) classes of non-linear continuous structural causal models with additive, multivariate Gaussian noise. (§3.2,3.3)

3. An algorithm that learns the parameters of the proposed causal model and disentangles single interventions from joint interventions. (§4)

4. An empirical validation of our method on both synthetic and real data.[3] (§5)

## 2  Related Work

**Disentangling multiple concurrent interventions:** [Parbhoo et al., 2021] study the question of disentangling multiple, simultaneously applied interventions from observational data. They propose a specially designed neural network for the problem and show good empirical performance on some datasets. However, they do not address the formal identification problem, nor do they address possible presence of unobserved confounders. By contrast our work derives the conditions under which identifiability holds. We moreover propose an algorithm that can disentangle multiple interventions even in the presence of unobserved confounders—as long as both observational and interventional data is available. Related work by [Parbhoo et al., 2020] investigated the intervention-disentanglement problem from a reinforcement learning perspective, where each intervention combination constitutes a different action that a reinforcement learning agent can take. Unlike this approach, our work explicitly focuses on modelling the interactions between interventions to learn their individual effects. Closer to our work is [Saengkyongam and Silva, 2020], who investigate identifiability of joint effects from observations and single-intervention data. They prove this is not generally possible, but provide an identification proof for non-linear causal models with additive Gaussian noise. Our work addresses a complementary question; we want to learn the effect of a single-intervention from observational data and sets of interventions. Additionally, another difference between our work and that of [Saengkyongam and Silva, 2020] is that they do not consider identification of individual-level causal effects given observed covariates. In a precursor to the work by [Saengkyongam and Silva, 2020], [Nandy et al., 2017] developed a method to estimate the effect of joint interventions from observational data when the causal structure is unknown. This approach assumed linear causal models with Gaussian noise, and only proved identifiability in this case under a sparsity constraint. However, like [Saengkyongam and Silva, 2020], our result does not need the linearity assumption, and no sparsity constraints are required in our identification proof. Finally, others including [Schwab et al., 2020, Egami and Imai, 2018, Lopez and Gutman, 2017, Ghassami et al., 2021] explored how to estimate causal effects of a single categorical-, or continuous-valued treatment, where different intervention values can produce different outcomes. Unlike our work, they do not consider multiple concurrent interventions that can interact.

**Combining observations and interventions:** [Bareinboim and Pearl, 2016] have investigated non-parametric identifiability of causal effects using both observational and interventional data, in a paradigm they call "data fusion." More general results were studied by [Lee et al., 2020], who provided necessary and sufficient graphical conditions for identifying causal effects from arbitrary combinations of observations and interventions. Recent work in [Correa et al., 2021] explored identification of counterfactual—as opposed to interventional—distributions from combinations of observational and interventional data. Finally, [Ilse et al., 2021] investigated the most efficient way to combine observational and interventional data to estimate causal effects. They demonstrated they could significantly reduce the number of interventional samples required to achieve a certain fit when

---

[3]The code to reproduce our results is available at github.com/olivierjeunen/disentangling-neurips-2022.

adding sufficient observational training samples. However, they only prove their method theoretically in the linear-Gaussian case. In the non-linear case, they parameterise their model using normalising flows and demonstrate their method empirically. They only consider estimating single-interventions, and do not deal with multiple, interacting interventions.

**Additive noise models:** While certain causal quantities may not be generally identifiable from observational and interventional data, by imposing restrictions on the structural functions underlying causal models, one can obtain *semi-parametric* identifiability results. One of the most common weak restrictions used in the causal inference community are additive noise models (ANMs), first studied in the context of causal discovery by [Hoyer et al., 2009]—and still widely used today [Rolland et al., 2022]. ANMs limit the form of the structural equations to be additive with respect to latent noise variables—but allow nonlinear interactions between causes. [Janzing et al., 2009] used ANMs to devise a method for inferring a latent confounder between two observed variables. This is otherwise not possible without additional assumptions on the underlying causal model. See [Lee and Spekkens, 2017], [Lee et al., 2019], and [Dhir and Lee, 2020] for an extension of this approach beyond ANMs. ANMs have also been employed by [Kilbertus et al., 2020] to investigate the sensitivity of counterfactual notions of fairness to the presence of unobserved confounding. Our work proves that in certain classes of ANMs, the effect of a single-intervention can be identified from observational data and sets of interventions—even in the presence of unobserved confounders. Moreover, we show how to incorporate observed covariates in these ANMs to learn the heterogeneous effects of single-interventions.

## 3   Model Identifiability

*Identifiability* is a fundamental concept in parametric statistics that relates to the quantities that can, or can not, be learned from data [Rothenberg, 1971]. An estimand is said to be identifiable from data if it is theoretically possible to learn this estimand, given infinite samples. If two causal models coincide on said data then they must coincide on the value of the estimand in question. Hence if one finds two causal models which agree on said data, but disagree on the estimand, then the estimand is not identifiable unless further restrictions are imposed. In this section, we provide identification proofs for single-variable interventional effects from observational data and joint interventions, for several model classes. Our theoretical analysis provides insights into the fundamental limitations of causal inference—and the assumptions that are required for identification.

**Problem Definition.**   We adopt the Structural Causal Model (SCM) framework as introduced by [Pearl, 2009]. An SCM $\mathcal{M}$ is defined by $\langle \{C, X, Y\}, U, f, \mathsf{P}_U \rangle$, where $\{C, X, Y\}$ are endogenous variables separated into covariates $C$, treatments $X$, and the outcome $Y$, $U$ are exogenous variables (possibly confounders), $f$ are structural equations, and $\mathsf{P}_U$ defines a joint probability distribution over the exogenous variables.

The SCM $\mathcal{M}$ also induces a causal graph—where vertices represent endogenous variables, and edges represent structural equations. Vertices with outgoing edges to an endogenous variable $X_i$ are denoted as the parent set of this variable, or $\mathrm{PA}(X_i)$. Typically, the observed covariates $C$ causally influence the treatments as well as the outcome, and are a part of this set. Every endogenous variable $X_i$ (including $Y$) is then a function of its parents in the graph $\mathrm{PA}(X_i)$ and a latent noise term $U_i$, denoting the influence of factors external to the model:

$$X_i := f_i(\mathrm{PA}(X_i), U_i). \tag{1}$$

In *Markovian* SCMs, these latent noise terms are all mutually independent. However, in general, distinct noise terms can be correlated according to some global distribution $\mathsf{P}_U$. In this case, such correlation is due to the presence of *unobserved confounders*.

An intervention on variable $X_i$ is denoted by $\mathrm{do}(X_i = x_i)$, and it corresponds to replacing its structural equation with a constant, or removing all incoming edges in the causal graph. The core question we wish to answer in this work, is under which conditions the treatment effect of a single intervention can be disentangled from joint interventions and observational data. That is, given samples from the data regimes that induce

$$\mathbb{E}[Y|X_i = x_i, X_j = x_j, C = c], \text{ and } \mathbb{E}[Y|\mathrm{do}(X_i = x_i, X_j = x_j), C = c],$$

when can we learn conditional average causal effects:

$$\mathbb{E}[Y|\mathrm{do}(X_i = x_i), X_j = x_j, C = c], \text{ or } \mathbb{E}[Y|X_i = x_i, \mathrm{do}(X_j = x_j), C = c]?$$

Table 2: Interventional distributions on $X_1$ under SCMs $\mathcal{M}$ and $\mathcal{M}'$.

| $\mathsf{P}_{\mathcal{M}}(Y, X_2\|do(X_1))$ | | $Y = 0$ | $Y = 1$ |
|---|---|---|---|
| $do(X_1 = 0)$ | $X_2 = 0$ | $1$ | $0$ |
| | $X_2 = 1$ | $0$ | $0$ |
| $do(X_1 = 1)$ | $X_2 = 0$ | $1 - p$ | $0$ |
| | $X_2 = 1$ | $0$ | $p$ |

| $\mathsf{P}_{\mathcal{M}'}(Y, X_2\|do(X_1))$ | | $Y = 0$ | $Y = 1$ |
|---|---|---|---|
| $do(X_1 = 0)$ | $X_2 = 0$ | $1 - p$ | $0$ |
| | $X_2 = 1$ | $p$ | $0$ |
| $do(X_1 = 1)$ | $X_2 = 0$ | $1 - p$ | $0$ |
| | $X_2 = 1$ | $0$ | $p$ |

In what follows, we first show that this quantity is not identifiable without restrictions on the causal model—a proof by counterexample. We then go on to prove, again by counterexample, that simply assuming ANMs without restrictions on the structure of the causal graph—the core assumption made by [Saengkyongam and Silva, 2020]—is not enough for this complementary research question. Finally, we prove identifiability of the treatment effect for ANMs without *causal* interactions between treatments. Note that this latter case does not mean treatments are independent: treatments can be influenced by observed covariates and unobserved confounders, and their interactions on the outcome are defined by the unrestricted function $f_Y$.

## 3.1 Unidentifiability for general SCMs

For simplicity, but without loss of generality, we consider two treatment variables $\{X_1, X_2\}$ and no covariates. We will show that two different SCMs $\mathcal{M}$ and $\mathcal{M}'$ can yield identical observational distributions, as well as joint interventional distributions. They will also agree on the single-variable interventional distribution

Table 1: SCMs for 3.1

| $\mathcal{M}$ | $\mathcal{M}'$ |
|---|---|
| $X_1 = U_1$ | $X_1 = U_1$ |
| $X_2 = X_1 U_2$ | $X_2 = U_2$ |
| $Y = X_1 X_2 U_Y$ | $Y = X_1 X_2 U_Y$ |

for treatment $X_2$, but disagree on the single-variable effect of treatment $X_1$. As such, given data from sets of interventions, this example shows the treatment effect of single-variable interventions is not generally identifiable. Our SCMs are defined in Table 1, where $U_1 = U_2 = U_Y \sim \text{Bernoulli}(p)$. The observational, joint, and single-variable interventional distributions on $X_2$ are identical under $\mathcal{M}$ and $\mathcal{M}'$; we defer them to the supplemental material. The interventional distribution on $X_1$ differs for $\mathcal{M}$ and $\mathcal{M}'$, as shown in Table 2. This counterexample shows that single-variable interventional effects are not identifiable for general SCMs, given observational data and joint interventions.

## 3.2 Unidentifiability for general ANMs

An Additive Noise Model is an SCM where the influence of the latent noise variables is restricted to be additive in the structural equations. That is, Eq. 1 is restricted to the form:

$$X_i := f_i(\text{PA}(X_i)) + U_i. \tag{2}$$

Following [Saengkyongam and Silva, 2020], we additionally assume the noise distribution to be a zero-centered Gaussian with an arbitrary covariance matrix: $\mathsf{P}_U \sim \mathcal{N}(0, \Sigma)$. Table 3 defines two SCMs that satisfy these assumptions. $\mathcal{M}$ and $\mathcal{M}'$ yield identical observational, joint interventional, and single-interventional distributions on $X_2$. However—they disagree on the causal effect of intervening on $X_1$. An intuitive underlying reason for this, is that we *can* identify the expression $\mathbb{E}[X_2|X_1 = x_1] = f_2(x_1) + \mathbb{E}[U_2|X_1 = x_1]$, but have no tools to disentangle the effects coming from the structural equation, $f_2(x_1)$, from those stemming from the additive noise, $\mathbb{E}[U_2|X_1 = x_1]$. As a result, $\mathbb{E}[X_2|do(X_1 = x_1)] = f_2(x_1)$ is not identifiable, proving that general ANMs with unrestricted causal structures are insufficient for disentangling treatment effects. In this example, and in general SCMs of this structure, joint interventions *can* be disentangled for the *consequence* treatment $X_2$, but not for the *causing* treatment $X_1$:

**Theorem 1** (Identifiability of disentangled conditional average treatment effects in additive noise models with a causal dependency between treatments).
*Let $\mathcal{M} = \langle \{\boldsymbol{C}, \boldsymbol{X}, Y\}, \boldsymbol{U}, \boldsymbol{f}, \mathsf{P}_U \rangle$ be an SCM, where*

$$X_i = f_i(\boldsymbol{C}) + U_i, \qquad X_j = f_j(\boldsymbol{C}, X_i) + U_j, \qquad Y = f_Y(\boldsymbol{C}, \boldsymbol{X}) + U_Y,$$

*and $\mathsf{P}_U \sim \mathcal{N}(0, \Sigma)$. The estimand $\mathbb{E}[Y|do(X_j), C]$ is identifiable from the conjunction of two data regimes: (1) the observational distribution, and (2) the joint interventional distribution on $(X_i, X_j)$.*

Table 3: SCMs for 3.2, where $(U_1, U_2, U_Y)_{\mathcal{M}} \sim \mathcal{N}(0, \Sigma)$ and $(U_1, U_2, U_Y)_{\mathcal{M}'} \sim \mathcal{N}(0, \Sigma')$.

| $\mathcal{M}$ | $\mathcal{M}'$ |
|---|---|
| $X_1 = U_1$ | $X_1 = U_1$ |
| $X_2 = X_1 + U_2$ | $X_2 = 2X_1 + U_2$ |
| $Y = X_1 X_2 + U_Y$ | $Y = X_1 X_2 + U_Y$ |

where $\Sigma = \begin{bmatrix} 1 & 1 & 1 \\ 1 & 1 & 1 \\ 1 & 1 & 1 \end{bmatrix}$, and $\Sigma' = \begin{bmatrix} 1 & 0 & 1 \\ 0 & 0 & 0 \\ 1 & 0 & 1 \end{bmatrix}$.

We prove this by showing that the structural equations are identifiable from the joint interventional regime—whereas the necessary (co-)variances are identifiable from observational data. Formal proofs are deferred to the supplementary material.

### 3.3 Identifiability for Symmetric ANMs

The crux of the problem of non-identifiability in Section 3.2 comes from the fact that treatment $X_1$ has a direct causal effect on treatment $X_2$. In many realistic applications, this might never occur. Treatments will decidedly be correlated, but this can be encoded in the SCM via either the observed covariates or the unobserved confounders. Then still, as we have no restrictions on the functional form of the structural equations with respect to the treatments (i.e. $f_Y$), treatments can still yield highly nonlinear interaction effects on the outcome. As such, we now focus on the case where the SCM is devoid of causal links between treatments, but includes observed covariates as well as unobserved confounders. In this case, the structural equations take the following form:

$$X_i = f_i(\boldsymbol{C}) + U_i, \qquad\qquad Y = f_Y(\boldsymbol{C}, \boldsymbol{X}) + U_Y. \qquad (3)$$

Identifiability in this setting requires additional assumptions about the correlation between the covariates and confounders. That is, without any assumptions on the nature of the relation between $C$ and $U$, our estimand is *not* identifiable. We include a proof (by counterexample) of this result in the supplemental material. Intuitively, the problem is that allowing $\mathbb{E}[U_Y|C]$ to be non-zero impacts the identifiability of $f_Y$ under joint interventions—as $f_Y$ also depends on $C$—whilst this is crucial to allow for disentanglement. As we have shown additional restrictions on the class of SCMs are necessary, we now assume independence between $C$ and $U$. Future work can explore whether this can be relaxed to allow for linear relationships, among others.

Fig. 1 visualises the structure of the causal graph we assume, when we have $K$ treatment variables. We call such causal graphs "symmetric". SCMs with this structure occur in many application areas from economics to online advertising and beyond. In SCMs of this form, the causal effect of all single-variable interventions can be disentangled from observational and joint interventional data:

**Theorem 2** (Identifiability of disentangled conditional average treatment effects in additive noise models with symmetric structure)**.**
*Let* $\mathcal{M} = \langle \{\boldsymbol{C}, \boldsymbol{X}, Y\}, \boldsymbol{U}, \boldsymbol{f}, \mathsf{P}_U \rangle$ *be an SCM, where*

$$X_i = f_i(\boldsymbol{C}) + U_i, \quad \forall i = 1, \dots, K,$$
$$Y = f_Y(\boldsymbol{C}, \boldsymbol{X}) + U_Y,$$

$C \perp\!\!\!\perp U$, *and* $\mathsf{P}_U \sim \mathcal{N}(0, \Sigma)$. *The estimand* $\mathbb{E}[Y|do(X_i), C]$ *is identifiable from the conjunction of two data regimes: (1) the observational distribution, and (2) any interventional distribution on a set of treatments* $\boldsymbol{X}_{int} \subseteq \boldsymbol{X}$ *that holds* $X_i$: $X_i \in \boldsymbol{X}_{int}$.

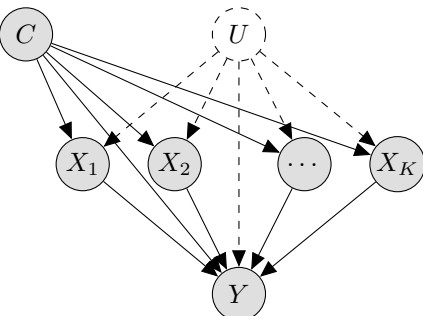

Figure 1: DAG for a symm. SCM with covariates and unobserved confounders.

In this section, we have studied for several classes of SCMs whether causal effects can be disentangled. Our results provide insights into the fundamental limits of learning and inference, and help crystallise which assumptions are necessary and sufficient to make disentanglement of effects from sets of interventions feasible. Figure 2 visualises and summarises our key results denoting *which* single-variable causal effects can be disentangled in ANMs with zero-mean Gaussian noise. In what follows, we present a learning methodology and validate our theoretical identifiability results with empirical observations on synthetic and real data.

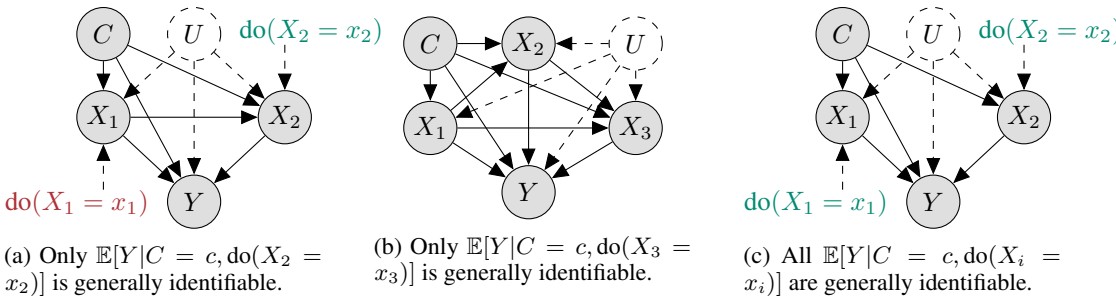

(a) Only $\mathbb{E}[Y|C = c, \mathrm{do}(X_2 = x_2)]$ is generally identifiable.

(b) Only $\mathbb{E}[Y|C = c, \mathrm{do}(X_3 = x_3)]$ is generally identifiable.

(c) All $\mathbb{E}[Y|C = c, \mathrm{do}(X_i = x_i)]$ are generally identifiable.

Figure 2: Causal DAGs illustrating under which conditions the single-variable causal effect on the outcome $\mathbb{E}[Y|C = c, \mathrm{do}(X_i = x_i)]$ is identifiable from observational and joint interventional data.

## 4 Methodology

In this section, we introduce our methodology for estimating SCMs and providing estimates for treatment effects under *any* set of interventions. Estimating an SCM from a combination of observational and interventional data boils down to (1) estimating the structural equations, and (2) estimating the noise distribution. We extend the Expectation-Maximisation-style iterative algorithm proposed by [Saengkyongam and Silva, 2020] to handle observed covariates, and to *disentangle* causal effects instead of *combining* them. The resulting method is not limited to learning single-variable causal effects, as the results from [Saengkyongam and Silva, 2020] allow us to extend these newly learned single-variable effects to sets of interventions. As a result, our method is able to generalise to sets of interventions that were never observed in the training data, with the only restriction that every variable that makes up the set was part of *some* intervention set in the training data.

Say we intervene on a subset of treatments $\boldsymbol{X}_{\mathrm{int}} \subseteq \boldsymbol{X}$, and $\boldsymbol{X}_{\mathrm{obs}} \equiv \boldsymbol{X} \setminus \boldsymbol{X}_{\mathrm{int}}$. In general, we can decompose a causal query with interventions on $\boldsymbol{X}_{\mathrm{int}}$ as follows:

$$\mathbb{E}[Y|\boldsymbol{C}; \mathrm{do}(\boldsymbol{X}_{\mathrm{int}}); \boldsymbol{X}_{\mathrm{obs}}] = f_Y(\boldsymbol{C}; \boldsymbol{X}) + \mathbb{E}[U_Y|\boldsymbol{X}_{\mathrm{obs}}]. \tag{4}$$

Samples consist of observed values for all endogenous variables. For convenience, we denote a sample by $\boldsymbol{x}$. Suppose we have data from $d$ different data regimes, corresponding to $d$ different sets of interventions (the empty set $\emptyset$ corresponds to the observational regime). The full dataset consists of samples and their corresponding interventions $\mathcal{D} = \{(\boldsymbol{X}_{\mathrm{int}}; \boldsymbol{x})\}$.

**Estimating the structural equations.** We parameterise the structural equations with $\theta$, denoted as $f(\cdot; \theta)$. The Gaussian likelihood with covariance matrix $\Sigma$ is denoted as $\mathsf{P}_U(\cdot; \Sigma)$.

The likelihood for a single endogenous variable $x_i$ is defined as $L(x_i; \theta, \Sigma) = \mathsf{P}_U(x_i - f_i(\mathrm{PA}(x_i); \theta); \Sigma)$. The likelihood for a sample $\boldsymbol{x}$ is defined as the product of the likelihoods for every endogenous variable that was *not intervened on* in that sample:

$$L(\boldsymbol{x}; \boldsymbol{X}_{\mathrm{int}}, \theta, \Sigma) = \prod_{x_i \in \boldsymbol{X} \setminus \boldsymbol{X}_{\mathrm{int}}} L(x_i; \theta, \Sigma). \tag{5}$$

---

**Algorithm 1** SCM Estimation for Symm. ANMs

1: **Input:** Dataset $\mathcal{D}$
2: **Output:** Parameter estimates $\widehat{\theta}, \widehat{\Sigma}$
3: Initialise $\widehat{\theta}$ and $\widehat{\Sigma}$
4: **while** not converged **do**
5:     // Solve for $\theta$ with fixed $\widehat{\Sigma}$
6:     Optimise log-likelihood in Eq. 6
7:     // Solve for $\Sigma$ with fixed $\widehat{\theta}$
8:     Estimate $\widehat{\Sigma}$ from $\widehat{U} = \boldsymbol{x} - \boldsymbol{f}(\boldsymbol{x}; \widehat{\theta})$
    **return** $\widehat{\theta}, \widehat{\Sigma}$

---

Naturally, the log-likelihood of the dataset is then:

$$\ell(\mathcal{D}; \theta, \Sigma) = \sum_{(\boldsymbol{X}_{\mathrm{int}}, \boldsymbol{x}) \in \mathcal{D}} \log L(\boldsymbol{x}; \boldsymbol{X}_{\mathrm{int}}, \theta, \Sigma). \tag{6}$$

In principle, any iterative optimisation procedure can be used to maximise Eq. 6 when we fix the covariance $\Sigma$. Typically, an appropriate method is chosen with respect to the model parameterisation.

**Estimating the noise distribution.** Following [Saengkyongam and Silva, 2020], we note that the maximum likelihood estimate for the covariance matrix of a multivariate normal can be directly computed from the sample. This allows for efficient closed-form computation of this step. Additionally, note that the assumption of zero-mean Gaussian noise simplifies the proof and yields an analytical solution for this step, but this is not a general limitation. Indeed—what matters is that we can estimate the conditional noise $\mathbb{E}[U_Y|\boldsymbol{X}_{\text{obs}}]$, which can in principle be estimated via a different model. Algorithm 1 shows the full procedure to estimate the parameters of a symmetric ANM. At inference time, Eq. 4 allows us to estimate the outcome under any—even unseen—set of interventions.

## 5 Empirical Validation

We now empirically validate the effectiveness of our method in estimating SCMs from observational and joint-interventional data, and assess the accuracy of the inferred outcomes under varying sets of interventions. We do this both on synthetic data and semi-synthetic data based on real data from a medical study. The research questions we wish to answer from empirical results, are the following:

**RQ1:** Are we able to accurately estimate the outcome under previously unseen sets of interventions?
**RQ2:** Are we able to accurately estimate the structural equations and the noise distribution?
**RQ3:** How does our method handle varying levels of unobserved confounding?

### 5.1 Synthetic experiment

First we adopt a simulation setup, which gives us the freedom to vary the *true* underlying SCM and observe performance differences among competing methods. The structural equations $f_i, f_Y$ in Eq. 3 are polynomials with second-order interactions to illustrate the effectiveness of the learning method even when treatment effects are highly non-linear, and the optimisation surface is highly non-convex. We use the well-known Adam optimiser in our experiments [Kingma and Ba, 2014]. As a baseline learning method we adopt a regression model that estimates the outcome from its direct treatments using pooled data from different regimes. Here, the presence of unobserved confounders severely hinders the method to provide accurate estimates under varying sets of interventions, further motivating the need for causal models.[4]

We let $|\boldsymbol{C}| = |\boldsymbol{X}| = 4$, yielding $\theta_i \in \mathbb{R}^{11} \quad \forall f_i$, and $\theta_y \in \mathbb{R}^{37}$ for the parameterisation of $f_y$. The parameters for the structural equations are randomly sampled from a uniform distribution over $[-2, +2]$. The covariance matrix $\Sigma \in \mathbb{R}^{5 \times 5}$ is uniformly sampled from a uniform distribution over $[-1, +1]$, and then ensured to be positive semi-definite through Ledoit-Wolf shrinkage [Ledoit and Wolf, 2004]. We repeat this process 10 times with varying random seeds and report confidence intervals over obtained measurements.

We vary the size of the training sample $|\mathcal{D}| \in \{2^i \forall i = 5, \ldots, 13\}$, where the set of intervened treatments $\boldsymbol{X}_{\text{int}}$ is one of $\{\emptyset; \{X_0, X_1\}; \{X_1, X_2\}; \{X_2, X_3\}; \}$. As such, no single-variable interventions and the majority of the possible sets of interventions are never observed. From this data, we estimate an SCM using the procedure laid out in Algorithm 1.

Then, for every possible $(2^{|\boldsymbol{X}|})$ set of interventions, we simulate $10\,000$ samples and estimate the outcome based on our estimated SCM using Eq. 4. We report the Mean Absolute Error (MAE) between our estimated outcome and the true outcome. Note that, as the true outcome is Gaussian, the optimal estimate is the location of that Gaussian, which will have an expected deviation of $\sqrt{\frac{2}{\pi}}\sigma$ where $\sigma$ is the standard deviation on the noise parameter $U_Y$.

**Estimating outcomes (RQ1).** Figure 3 visualises the results from the procedure laid out above, increasing the size of the training sample over the x-axis and reporting the MAE on the y-axis. Note that both axes are logarithmically scaled, and the shaded regions indicate 95% confidence intervals. The plots clearly indicate that our method is able to provide accurate and close to optimal estimates for the outcome, under *any* possible set of interventions. Indeed, the plots labelled *observational*, $do(X_0, X_1)$, $do(X_1, X_2)$ and $do(X_2, X_3)$ show the accuracy of our method to estimate the outcome under interventions that occurred in the training data—but the remaining twelve plots relate to previously *unseen* data regimes. While our results allow us to *disentangle* joint interventions, the

---

[4]The code to reproduce our results is available at github.com/olivierjeunen/disentangling-neurips-2022.

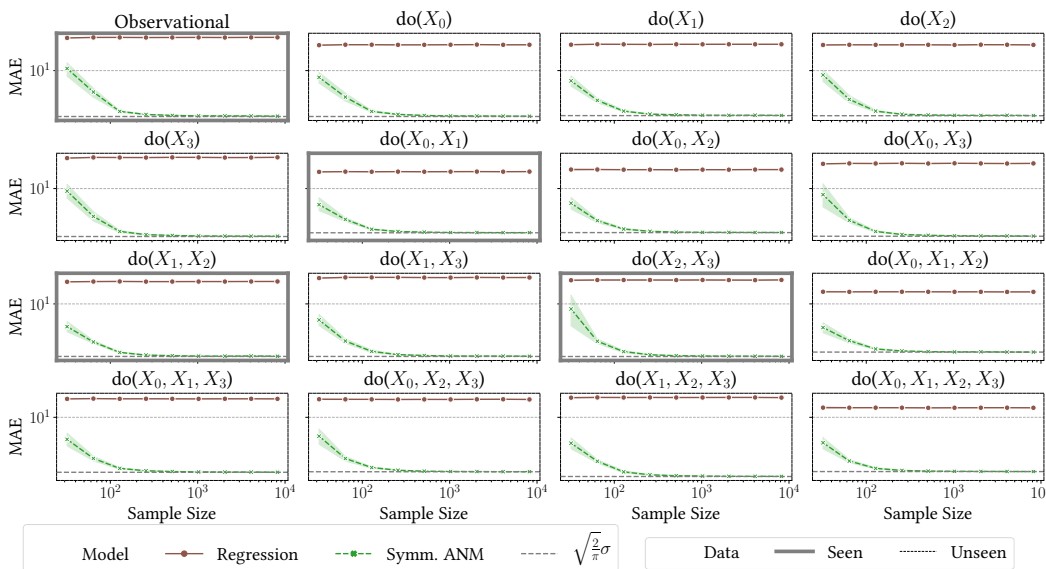

Figure 3: Mean Absolute Error for estimating outcome under varying sets of interventions, for varying sample sizes, comparing baseline regression with our method. We empirically show our estimator is unbiased and consistent, exhibiting optimal predictions even at modest training sample sizes.

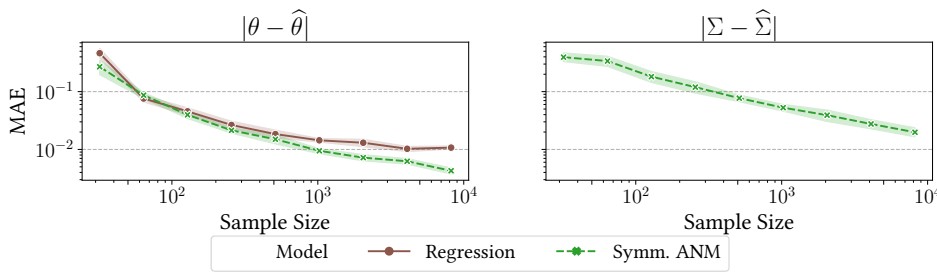

Figure 4: Mean Absolute Error for estimating the parameters of the underlying SCM, for varying sample sizes, comparing baseline regression with our method. We empirically validate our estimator is unbiased and consistent, giving accurate estimates even at modest training sample sizes.

results of [Saengkyongam and Silva, 2020] allow us to *combine* interventions. As such, our method incorporates (and subsumes) theirs, in order to generalise to arbitrary sets of interventions. In contrast, the regression method that is oblivious to confounders fails to accurately estimate the outcome under any data regime—even those that are seen in the training data, or those where confounders yield no influence on the outcome (i.e. all treatments are intervened on). As such, the reported results validate that the proposed method provides an unbiased and consistent estimator for the disentangled causal effect, learned from sets of interventions in the presence of unobserved confounders.

**Estimating SCMs (RQ2).** We visualise the results with respect to the accuracy of the parameter estimates in Figure 4. These measurements are obtained using the same procedure and runs as for RQ1. Increasing the size of the training sample over the x-axis, the leftmost plot shows the accuracy of the estimated parameters for $f_Y$. The rightmost plot shows the accuracy of the estimated covariance matrix for the multivariate normal that defines the noise distribution. As the baseline regression method is oblivious to the noise distribution, it is not included in the latter plot. Both axes are logarithmically scaled, and the shaded regions indicate 95% confidence intervals. We observe that our method is able to efficiently and effectively estimate the underlying causal model—empirically validating the identifiability results presented in this work.

## 5.2 Semi-synthetic experiment based on real medical data

We now show our method can accurately estimate unseen sets of interventions (**RQ1**) and demonstrate it is consistent under varying levels of unobserved confounding (**RQ3**) using a semi-synthetic setup on real-world data from the International Stroke Trial database [Carolei, 1997]. This was a large, randomized trial of up to 14 days of antithrombotic therapy after stroke onset. There are two possible treatments: the aspirin allocation dosage $\alpha_a$, and the heparin allocation dosage $\alpha_h$. The goal is to understand the effects of these treatments on a composite outcome, a continuous value in $[0, 1]$ predicting the likelihood of patients' recovery.

We partially follow the semi-synthetic setup laid out in Appendix 3 of [Zhang and Bareinboim, 2021]. Specifically, we adopt their probability table for the joint observational distribution of the covariates: the gender, age and conscious state of a patient. This table was computed from the dataset to reflect a real-world observational distribution. While [Zhang and Bareinboim, 2021] deal with single discrete treatments, we extend to the continuous and multi-treatment case where treatments can be interpreted as varying dosages of aspirin and heparin. This procedure is described in detail in the Supplementary Material.

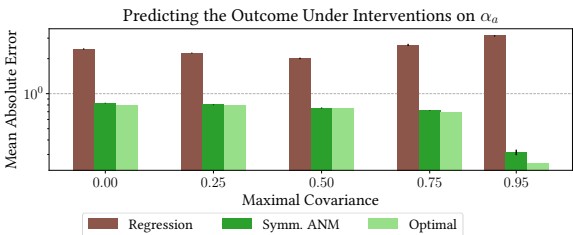

Figure 5: Mean Absolute Error for estimating the impact of interventions on treatment $\alpha_a$, learned only from observational and joint interventional data. We vary the magnitude of the covariance between treatments and outcome to assess the effect of varying confounding.

We add zero-mean Gaussian noise to the treatments and the outcome, where we randomly generate positive semi-definite covariance matrices with bounded non-diagonal entries—varying the limit on the size of the covariances in order to assess the effect of varying confounding on our method. We repeat this process 5 times, and sample 512 observational and 512 joint-interventional samples (both $\alpha_a$ and $\alpha_h$ to learn the SCM from). We evaluate our learning methods on 5 000 evaluation samples to predict the outcome under a single-variable intervention on the aspirin dose $\alpha_a$.

Figure 5 shows results with respect to the accuracy of the learnt model for making predictions on the outcome under varying single-variable interventions on the aspirin dosage $\alpha_a$. Naturally, as the regression method has no way of accounting for the confounding effect, it fails dramatically. Our proposed method is able to consistently approximate the optimal performance—which is attained by predicting the expectation of the outcome distribution and limited by its variance. For increasing levels of confounding, the gap between the optimal performing oracle and our learnt model increases slightly. As such, this semi-synthetic experiment validates the effectiveness of our proposed learning method, which can predict outcomes under unseen sets of interventions (**RQ1**) and is robust to varying levels of confounding (**RQ3**), for a use-case based on real-world medical data.

## 6 Conclusions & Future Work

In this work, we motivated the need for methods that can disentangle the effects of single interventions from jointly applied interventions. As multiple interventions can interact in possibly complex ways, this is a challenging task; even more so in the presence of unobserved confounders. First, we proved that such disentanglement is not possible in the general setting, even when we restrict the influence of the unobserved confounders to be additive in nature. By restricting the structure of the causal graph to be symmetric we showed identifiability *can* be acquired. Additionally, we showed how to incorporate observed covariates, and have empirically demonstrated our method.

Future work will tackle the case where the noise distribution is not restricted to a zero-mean multivariate Gaussian. As mentioned in Section 4, this assumption provides a closed-form expression for the conditional expectation of the outcome noise given the observed treatments—but universal function approximators could be used to obtain similar guarantees in more general model classes. Investigating whether the assumption of causal independence between the unobserved confounders and covariates can be relaxed, to allow for linear relationships say, provides another avenue to explore.

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
