# OpenReview forum: "Disentangling Causal Effects from Sets of Interventions in the Presence of Unobserved Confounders"
_NeurIPS.cc/2022/Conference — NeurIPS 2022 Accept_

### Official Review · Reviewer_P8j4 · 2022-06-20

**Rating:** 5
**Confidence:** 4
**Soundness:** 3 good
**Presentation:** 3 good
**Contribution:** 2 fair

**Summary:**

In this paper, the authors aim to disentangle the single intervention's effect with the observational data and interventional data upon **many variables** simultaneously, in the presence of latent confounders. The authors suppose the additive noise model, and propose two cases when the effect can be disentangled, one is the case when all the treatment variables are in exact causal order and we want to identify the causal effect of the treatment variable at the last of the order on the outcome, in Theorem 1, the other is the case when there are no causal relations between the treatment variables, in Theorem 2. Further, the authors present the method to disentangle the effect by EM algorithm.


**Questions:**

I am not quite sure whether the examples in the first paragraph are convincing. For example, people are targeted by multiple concurrent campaigns in computational advertising, or patients that possess many commodities may have to be simultaneously treated with multiple prescriptions. In these cases, our goal is usually to know how to intervene to make the people buy our commodity, or how to do to make patients restored from sickness. Why do we identify the single intervention's effect?

**Limitations:**

The authors should make it clear that it is an assumption that covariates $C$ are not allowed to be caused by the treatment variable.

**Strengths And Weaknesses:**

Strengths:

(+) The problem of disentangling single intervention's effect is interesting.

(+) In total, the paper is clearly written.

Weaknesses:

(-) According to my understanding, to the problem of disentangling single intervention's effect, the most interesting part is not to give some cases when the disentangling can be finished, but when the disentangling cannot be finished. A sufficient and necessary condition for the ANM model is interesting and novel. The current paper only presents two cases when the single intervention's effect can be identified. A deeper study is expected.


(-) In this paper, it is not allowed that some covariate $C$ is caused by the treatment variable, as Line 121. I think the authors should make it clear that it is an assumption in this paper. In general SCM framework, there are not such assumptions that the causal directions between covariates and treatments are from covariates to treatments.

(-) The formalization of the problem on Line 133 is not quite good. The intervention can be upon more than two variables, but in the format here there are only two intervention variables.

(-) I think it will be better if the authors can further clarify the motivation of the disentangling from interventions upon many variables. In the view of theory, Lee et al. 2020 have given the sound and complete result for the non-parametric identifiability with arbitrary experiments. The manuscript can be seen as a special case when we (1) consider ANM; (2) assume covariates cause both treatment and outcome; (3) Gaussian noise. I am not saying it is not good, I can understand that by this way there are some advantages in computation. But considering that the theoretical results in the manuscript are not complete, it is vital to argue for the importance of their results or methods in reality.

---

> ### Author Response · Authors · 2022-08-02
> **Response to Reviewer P8j4**
>
> Dear Reviewer,
>
> We thank you for your thorough evaluation of our work and your suggestions.
>
> You mention it would be interesting to provide cases where disentanglement cannot be done. This is exactly what we focus on in Sections 3.1, 3.2, paragraph 1 of 3.3, Figure 2, and Appendices C, D. Specifically, we show that disentanglement of single-variable effects is not guaranteed for general SCMs, and even not for general ANMs with multivariate Gaussian noise. We then consider which treatments’ causal effects can be disentangled when causal dependencies among treatments are present, as visualised in Figure 2.
> This structure builds up from the most general setting (where disentanglement is non-identifiable), incrementally adding in common assumptions (providing partial identifiability results along the way) until we have a full identifiability result.
> We welcome any specific suggestions the reviewer might have for further conditions to consider.
>
> You mention that in our framework, covariates cannot be caused by treatments. We agree that adding a line in the paper stating this assumption would help to clarify.
> Nevertheless, changing the direction of the causal arrow between $X$ and $C$ (ceteris paribus) does not change our identifiability result.
> Indeed, if the treatments cause the covariates, then the covariates are mediators that do not bias the association between treatment and outcome. That is, they are no longer confounders that need to be controlled for. As such, the conditional average treatment effect remains identifiable, and reduces to the average treatment effect (as there is nothing to condition on in this case).
> We thank the reviewer for pointing this out, and hope the reviewer agrees this in no way invalidates the core contributions of our work.
>
> You mention that the formalisation of the problem in Line 133 is incomplete, as it deals with specific random variables instead of sets. The goal of this example is merely to give a clear example with minimal notational clutter — but we can replace the random variables $X_i, X_j$ with sets to provide a general formalisation.
>
> Next, you mention that we should further clarify the motivation for disentanglement of single-variable treatment effects. In particular, you are not convinced that the causal effect of a single advertising campaign or a single prescription would be of interest. We will focus on these two examples:
> (1) in the advertising case, consider an online platform (retail, streaming,...) that runs multiple concurrent campaigns, targeting overlapping sets of users. Such a platform might want to consider the effects of certain types of advertising campaigns on user retention. This effect can be neutral for a set of interventions, because it might be offset by some individual positive and negative interventions. Nevertheless, some ad campaigns might have a positive effect and some ad campaigns might have a negative effect, and we wish to learn for every campaign what that effect is. These insights are crucial for the platform, to make decisions about future advertising campaigns they might want to run.
> (2) In the multiple prescription case, you mention that the goal is to make people recover, and not to understand the causal effects of single treatments. We believe that these go hand-in-hand. Indeed, dosages of multiple treatments can interact in non-linear ways on the outcome, and in order to identify the minimal set of interventions that maximises the positive effect on the outcome, we wish to disentangle the joint effects. This type of set-up is consistent with our semi-synthetic experiment in Section 5.2, where we disentangle the effects of the aspirin allocation dosage and the heparin allocation dosage on the likelihood of patients’ recovery.
>
> Finally, you mention the work of Lee et al. 2020, which we assume to be the following paper: http://proceedings.mlr.press/v115/lee20b/lee20b.pdf. The authors of that work provide necessary and sufficient *graphical* conditions for deciding whether a causal query is identifiable. The general case of disentangling the effects of multiple treatments is not identifiable from graphical conditions alone – as we show in our paper. Hence, a natural question to ask is under what semi-parametric assumptions does this causal query (i.e. the conditional average treatment effect given single variable interventions) become identifiable given access to observations and joint interventions. As discussed above, this is an important question. Hence it is important to explore under what assumptions this causal quantity can be identified and estimated, which is what our paper is concerned with.
>
> We appreciate the reviewer’s time and effort, and hope to have clarified the reviewer’s questions and adequately responded to their suggestions.
>
> Yours sincerely,
> The Authors

---

> > ### Comment · Reviewer_P8j4 · 2022-08-05
> > **Response**
> >
> > Dear authors:
> >
> >   Thank you for your response. It addresses my main questions. I will change my score.
> >
> > Best Regards,

---

### Official Review · Reviewer_sK28 · 2022-07-10

**Rating:** 6
**Confidence:** 3
**Soundness:** 3 good
**Presentation:** 3 good
**Contribution:** 3 good

**Summary:**

Assuming $C \perp U$ and $X_i \perp X_j \mid C, U$ ($i \not = j$) in Figure 2(c), this paper provides identification proofs under non-linear continuous structural causal models with additive, multivariate Gaussian noise. The estimation is identifiable from the conjunction of two data regimes: (1) the observational distribution, and (2) any intervention distribution on a set of treatments. Then, this paper extends the Expectation-Maximization-style iterative algorithm to disentangle causal effects.

**Questions:**

1. This paper claims that disentangling the effects of single interventions from jointly applied interventions is a challenging task. Thus, they assume $C \perp U$ and $X_i \perp X_j \mid C, U$ ($i \not = j$) in Figure 2(c). In this setting, is it still a challenge to disentangle the effects of single interventions?

2. Under additive multivariate Gaussian noise assumption, can we use traditional methods for single treatment analysis? That is, we can focus on a single treatment variable $X_i$ and treat other treatments $\{X_j\}_{j \not= i}$ as confounders.

3. Does this paper require prior knowledge of the structural equations?  The reasons for uniform distribution over [-2,+2] and [-1,+1] are?

4. The estimation is identifiable from the observational distribution. Why would this paper identifies causal effects from intervention distribution? To be honest, the acquisition of experimental data is also a problem.

5. As the baseline for this paper, I think some causal algorithms for distribution modeling should also be considered, such as GANITE (Yoon et al., 2018), SCIGAN (Bica et al. 2020), CEVAE (Louizos et al., 2017), and Intact-VAE (Pengzhou et al., 2022)...

6. In the presence of unmeasured confounders (unconfoundedness assumption is violated), can the causal effect of a single intervention variable be accurately estimated under non-linear continuous structural causal models with additive, multivariate Gaussian noise?

**Limitations:**

I do not foresee any major limitations and/or societal impacts.

**Strengths And Weaknesses:**

This paper claims that disentangling the effects of single interventions from jointly applied interventions is a challenging task. Thus, they assume $C \perp U$ and $X_i \perp X_j \mid C, U$ ($i \not = j$) in Figure 2(c). In this setting, is it still a challenge to disentangle the effects of single interventions? I will adjust my score according to the authors’ responses to my comments.

**Strengths**
- This paper considers a interesting problem in Multiple Treatment Effect and provide identification assumption.
- Theoretically, this paper provide that learning the effect of a single-intervention from both observational data and sets of interventions is not generally possible, but provide identification proofs demonstrating that it can be achieved under non-linear continuous structural causal models with additive, multivariate Gaussian noise.

**Weaknesses**
- Additive multivariate Gaussian noise assumption is not a general condition, which is rarely satisfied in practice.
- In Experiment Sectoin, the regression method is not a representative baseline in causal inference.
- In the observed data, disentangling single-treatment effect does not seem to be a challenge. We can analyze the single-treatment effect one by one and treat all other variables as confounders.

-----------------------
I thank the authors for addressing my comments. I increase my score to 6.

---

> ### Author Response · Authors · 2022-08-02
> **Response to Reviewer sK28**
>
> Dear Reviewer,
>
> We thank you for your thorough evaluation of our work and your suggestions.
>
> We agree that the multivariate normal assumption will not always be met. Nevertheless, our results show that some assumptions are necessary. We discuss this in lines 247–254: the multivariate Gaussian assumption allows us to obtain an analytical form for $\mathbb{E}[U_y | X_{\rm obs}]$, but in practice this quantity can be estimated by any other model. Universal function approximators can possibly lead to identifiability results in these more general settings.
>
> Moreover, our results show that even under the multivariate Gaussian assumption (and without further assumptions on the causal structure among treatments, and independence between covariates and confounders), single-variable treatment effects are _not_ identifiable (paragraph 1 of 3.3, Appendix D). As such, we do not believe this assumption to be overly restrictive. Lastly, we would like to point out that the multivariate Gaussian ANM assumption is common in recent related work (Rolland et al. 2022, Kilbertus et al. 2020, Saengkyongam and Silva, 2020).
>
> You mention “In the observed data, disentangling single-treatment effect does not seem to be a challenge”, and “the estimation is identifiable from the observational distribution”.
> We do not agree with this statement. All three of our proofs by counterexample assume both access to observational data as well as data containing joint interventions. In all three, the causal effect of single-variable interventions is not identifiable. As such, it is clearly not identifiable from observational data alone, without any further assumptions.
>
> We do not require prior knowledge of the structural equations – the uniform distributions are merely used to generate random SCMs, which our proposed method can learn the parameters and noise distribution for.
>
> You mention the acquisition of experimental data is a problem. We believe our method helps mitigate the cost of acquiring experimental data — as it foregos the need to run additional experiments with single-treatment interventions. Indeed, under the right conditions, these can be identified from observational data and joint interventional data. Furthermore, the cost of acquiring experimental data largely depends on the use-case. As an example of settings where interventional data will be abundant, typical web platforms run large-scale interventional studies (such as A/B-tests) continually (but treatment groups will overlap, obfuscating single-treatment causal effects).
>
> You mention that we could consider recently proposed general algorithms for causal modelling.
> Each of the proposed methods is conceptually optimising the same thing as standard regression adjustment given the observed confounders/covariates. Indeed, GANITE, SCIGAN and $\beta$-Intact-VAE make an _unconfoundedness_ assumption that we do _not_ rely on in this work; whereas CEVAE assumes access to proxy variables to the confounders (which we do not).
> Hence, as they can't account for unobserved confounders (and thus the interactions between treatments) they will suffer similarly to standard regression. Indeed, the main usefulness of algorithms such as GANITE is that they can control for observed confounders more efficiently/accurately than regression when there are large numbers of observed confounders. As our experiments are not concerned with that regime, we will not gain much by comparing to such algorithms. We do agree that there is an interesting opportunity for future work in combining our EM-style approach with stronger base learners to approximate the structural equations.
> Nevertheless, we doubt the added value a comparison would bring to the current paper without diluting the core contributions of our work. The main scientific contribution of our work focuses on identifiability conditions for disentangling joint effects, and our empirical results corroborate our theoretical findings. An in-depth study of the efficacy of existing modelling algorithms is out-of-scope for the purposes of this paper.
>
> Lastly, you ask whether we still have identifiability when _“the unconfoundedness assumption is violated”_ . We would like to stress that we have never made such an unconfoundedness assumption: our work provides identifiability results when unobserved confounders are present – indeed, this is reflected in the title of the paper.
>
> As Reviewer 5byu points out — we could additionally make a (partial) unconfoundedness assumption in order to relax other assumptions on the DAG structure (and causal independence among treatments, specifically).
>
> We appreciate the reviewer’s time and effort, and hope to have clarified the reviewer’s questions and adequately responded to their suggestions.
>
> Yours sincerely,
> The Authors

---

### Official Review · Reviewer_FYX4 · 2022-07-11

**Rating:** 6
**Confidence:** 2
**Soundness:** 3 good
**Presentation:** 3 good
**Contribution:** 3 good

**Summary:**

The paper considers estimating the causal effect of a single intervention when multiple interventions are applied at the same time.
The paper characterizes several cases where the single intervention's effect can be identified.
In particular, the paper considers the identifiability characterization under non-linear structural models, unobserved confounders, and interactions among interventions. The paper also designs an algorithm that learns the causal effect by combining observed data and interventional data.


**Questions:**

* It would be very helpful to discuss the necessity and sufficiency of the identifiability conditions.

* It would be very helpful to provide some theoretical insights of the proposed algorithm.

* It would be very helpful to discuss how the identifiability conditions could be verified/checked on a real dataset.


**Limitations:**

Yes.

**Strengths And Weaknesses:**

Strengths:

* The paper extends previous works of linear structural models to allow non-linear dependence. The paper also takes into account unobserved confounders---a fundamental challenge in causal inference.

* The paper employs observational data to augment interventional data. This is impactful because there are usually more observational datasets available, and often in large scale.

Weaknesses:

* The identifiability discussed in the paper relies on the multivariate normal assumption. This may not always be satisfied in practice. For example, some variables may be binary and do not follow Gaussian distributions.

---

> ### Author Response · Authors · 2022-08-02
> **Response to Reviewer FYX4**
>
> Dear Reviewer,
>
> We thank you for your thorough evaluation of our work and your suggestions.
>
> We agree that the multivariate normal assumption will not always be met. Nevertheless, our results show that some assumptions are necessary. We discuss this in lines 247–254: the multivariate Gaussian assumption allows us to obtain an analytical form for $\mathbb{E}[U_y | X_{\rm obs}]$, but in practice this quantity can be estimated by any other model. Universal function approximators can possibly lead to identifiability results in these more general settings.
>
> Moreover, our results show that even under the multivariate Gaussian assumption (and without further assumptions on the causal structure among treatments, and independence between covariates and confounders), single-variable treatment effects are _not_ identifiable (paragraph 1 of 3.3, Appendix D). As such, we do not believe this assumption to be overly restrictive. Lastly, we would like to point out that the multivariate Gaussian ANM assumption is common in recent related work (Rolland et al. 2022, Kilbertus et al. 2020, Saengkyongam and Silva, 2020).
>
> You mention it would be helpful to discuss the necessity and sufficiency of the identifiability conditions. We believe that the sufficiency of the conditions is shown by our identifiability proofs. We agree that our proofs by counterexample for un-identifiability without these conditions do not imply that _our_ conditions are necessary – merely that _some_ conditions are necessary.
> We welcome any specific suggestions for other conditions to consider in future work.
>
> You mention it would be helpful to provide theoretical insights into the proposed algorithm. As we mention in the paper, it is essentially an Expectation-Maximisation (EM) algorithm where each step either maximises the log-likelihood with respect to a fixed covariance matrix, or fits the covariance matrix of the unobserved noise terms given parameters of the structural equations that relate observations to those noise terms. Theoretical properties of this iterative optimisation procedure will be closely related to those known for the EM algorithm. We showed in our proof that there are closed form expressions for all the noise term covariances, so that step is computationally trivial. Hence the computational complexity resides in the log-likelihood optimisation step and the number of iterations the EM algorithm requires.
>
> Identifiability assumptions usually cannot be checked empirically — this is not a specific limitation to the assumptions we adopt in this work, but a general limitation of theoretical causal inference research. A prominent example of an assumption that cannot be empirically validated is the often-made “unconfoundedness” assumption (which we do not make in this paper).
>
> We appreciate the reviewer’s time and effort, and hope to have clarified the reviewer’s questions and adequately responded to their suggestions.
>
> Yours sincerely,
> The Authors

---

> > ### Comment · Reviewer_FYX4 · 2022-08-09
> > **Reply to the authors' response**
> >
> > Thank you for the response! I found the discussion of the multivariate Gaussian assumption very helpful.
> >
> > A quick follow-up of the optimization algorithm. It appears to me that the procedure iteratively optimizes $\\theta$ and $\\Sigma$. I am not fully aware where is the E step, and it would be very helpful to clarify it.

---

> > > ### Author Response · Authors · 2022-08-09
> > > **Response to Reviewer FYX4's comment**
> > >
> > > Dear Reviewer,
> > >
> > > Thank you for acknowledging our rebuttal — we intend to include this expanded discussion and intuition for the multivariate Gaussian assumption in the final version of the manuscript.
> > >
> > > In our analogy to the classical EM algorithm, the "Expectation" step corresponds to computing the Maximum Likelihood Estimate for $\widehat{\Sigma}$ (the covariance matrix for the latent noise variables), whilst keeping $\widehat{\theta}$ fixed (the estimated parameters for the structural equations).
> > > The "Maximisation" step then subsequently aims to find the parameters $\widehat{\theta}$ that maximise the expected log-likelihood (given a fixed distribution for the unobserved confounders computed in the preceding E-step: $\widehat{\Sigma}$).
> > >
> > > Indeed, the EM algorithm is an iterative estimation method that provides a way to deal with unobserved latent variables, which are the confounders in our problem setting.
> > >
> > > Yours sincerely,
> > > The Authors

---

> > > > ### Comment · Reviewer_FYX4 · 2022-08-09
> > > > **Response to the EM question**
> > > >
> > > > Thanks for the reply! May I understand the estimation of $\\Sigma$ and $\\theta$ as the M-step, and the computation of the likelihood expectation given the current $\\Sigma$ and $\\theta$ as the E-step?

---

> > > > > ### Author Response · Authors · 2022-08-09
> > > > > **Response to the EM Question**
> > > > >
> > > > > Not exactly.
> > > > >
> > > > > The estimation of $\Sigma$ comprises the E-step: we use the training sample to estimate the distribution for the latent variables, which we can then use to compute the expected likelihood.
> > > > > Because of the additive noise assumption, we can recover estimates for the noise terms themselves (i.e. $\widehat{U} = x - f(x; \widehat{\theta})$, as shown in Line 8 of Algorithm 1, p6).
> > > > > The MLE for $\widehat{\Sigma}$ is then given by the sample covariance matrix using the estimated $\widehat{U}$ — this is highlighted in the manuscript at L247-249.
> > > > > The final part of the E-step is then to plug this $\widehat{\Sigma}$ into the expression for the likelihood, given at L238-244.
> > > > >
> > > > >
> > > > > The optimisation of $\theta$, using the fixed $\widehat{\Sigma}$ and thus the expression for the log-likelihood that we have obtained in the E-step, is then the M-step. Indeed, this latter step learns $\theta$ to *maximise* the likelihood that is the output of the E-step. In our experiments, we obtain the $\widehat{\theta}$ estimates using the Adam optimiser (see L268). These $\widehat{\theta}$  estimates are then used in the subsequent E-step to obtain updated estimates of the noise terms $\widehat{U}$, and the iterative procedure continues.

---

### Official Review · Reviewer_5byu · 2022-07-12

**Rating:** 6
**Confidence:** 4
**Soundness:** 3 good
**Presentation:** 4 excellent
**Contribution:** 3 good

**Summary:**

This paper is on disentangling the effects of single interventions from jointly applied interventions, given observational data and the joint interventions. In particular, this paper shows that the causal effect of single interventions is generally unidentifiable, but it is identifiable for the so-called symmetric ANMs, where there is no direct causal connection among the treatment variables. Moreover, this paper proposes a practical method to estimate the causal effect of single interventions.

**Questions:**

If only some of the treatment variables are influenced by hidden confounders, then what will the identifiability conditions be? Can they become weaker?

**Limitations:**

The authors did not discuss the limitations and potential negative societal impact of their work.

**Strengths And Weaknesses:**

Strengths:
The motivation is clearly stated at the beginning of the paper, and the overall organization is good. The studied problem is new to me.

The paper gives both theoretical identifiability results and practical estimation methods.

Weaknesses:
The authors give the identifiability in rather strong assumptions. One is that if there is no direct causal connection among the treatment variables for ANMs, then the causal effect of single interventions is identifiable. The other is that for general ANMs, the causal effect of the sink treatment variable is identifiable. It could be interesting to investigate weaker identifiability conditions.

In addition, in some cases, we may want to know the identifiability of a subset of the interventions, so it would be also interesting to see the identifiability conditions for the causal effect of a subset of interventions.

---

> ### Author Response · Authors · 2022-08-02
> **Response to Reviewer 5byu**
>
> Dear Reviewer,
>
> We thank you for your thorough evaluation of our work and your suggestions.
>
> You mention that it would be interesting to investigate weaker identifiability conditions. In particular, you ask whether other conditions can be lifted when we assume that some treatment variables are not influenced by confounding. This would indeed be another way to solve the problem in our un-identifiability proof by counterexample in Appendix D.
> In the paper, we make assumptions about the structural equations to pin down the noise distribution. As you rightly point out, we can reverse this line of thought, making assumptions about the noise distribution to pin down the structural equations.
>
> Assume a treatment $X_i$ is not influenced by the hidden confounders. As $\mathbb{E}[Y|X_i=x_i; C=c] = \mathbb{E}[Y|{\rm do}(X_i=x_i); C=c]$ (because $\mathbb{E}[U_y|X_i=x_i; C=c]=\mathbb{E}[U_y]=0$), the single-treatment causal effect of intervening on treatment $X_i$ is trivially identifiable from observational data.
> Our DAG structure in Figure 1 implies we can reframe the problem with $X_i$ in the set of covariates instead of the set of treatments. This reinterpretation shows that $X_i$ can then causally influence all other treatments (as all other covariates do), while retaining identifiability of single-variable interventional effects for all treatments other than $X_i$ (as proven by Theorem 2).
>
> Nevertheless, as we show in Appendix D, assuming neither independence between the treatments and confounders nor causal independence among treatments does not yield identifiability in the general case. We welcome any specific suggestions for other weaker conditions to consider in future work.
>
> You also ask what the identifiability conditions would be for the causal effect of a subset of interventions.
> Our theoretical results show how single-variable effects can be identified from observational and joint interventional data. This can be merged with the results presented by Saengkyongam and Silva (2020), which in turn combine single-variable interventions to newly formed sets of interventions. As we mention on line 224 and show in Figure 3, this allows us to identify causal effects for previously unseen sets of interventions.
>
> We agree that the question of identifiability of the causal effect of intervening on a general subset, without the identifiability of causal effects of single-variable interventions given in this work, is an interesting open research question.
>
> We appreciate the reviewer’s time and effort, and hope to have clarified the reviewer’s questions and adequately responded to their suggestions.
>
> Yours sincerely,
> The Authors

---

> > ### Comment · Reviewer_5byu · 2022-08-09
> > **Thank you**
> >
> > Thank you for the response. I agree with the authors that identifiability with weaker conditions is an open research question.

---

### Author Response · Authors · 2022-08-09
**Final response to Reviewers and (Senior) Area Chairs**

Dear Reviewers and (Senior) Area Chairs,

We appreciate the explicit acknowledgement of our rebuttal, which led both reviewers sK28 and P8j4 to increase their scores. All reviewers now recommend acceptance.

We trust that our rebuttal and further responses addressed any outstanding questions and suggestions that reviewers 5byu and FYX4 had.

If anything remains unclear, we would be happy to use the remainder of the discussion period to clarify.

Otherwise, we would like to thank you all again for your time and effort.

Yours sincerely,
The Authors

---

### Meta-Review · Area_Chair_K5bA · 2022-08-25

**Recommendation:** Accept
**Confidence:** Less certain

**Metareview:**

The authors describe a method to estimate the effect of an intervention on a single variable X_j in a setting where data from interventions on multiple variables is available. This data is combined with observational data to arrive at an identification formula. The procedure relies heavily on (a) assuming that the noise / confounding is additive (b) that there is no causal relationship between the covariates and to lesser extent on (c) a Gaussianity assumption. The identification strategy is based on a neat idea, and it is refreshing to read about novel identification results that make use of both observational and interventional data. However, it is unclear how robust the procedure is under small violations of the assumptions. Furthermore, in many applications, some of the covariates might be categorical. This is problematic under the additive confounding assumption. To summarize, this paper presents an interesting idea that allows combining evidence from observational and interventional data. In its current form, the setting is likely too artificial to be directly relevant for practice.

**Award:**

No

---

### Decision · Program_Chairs · 2022-09-14

Accept